# A New Class of Strongly Asymmetric PKA Algorithms: SAA-5

**Luigi Accardi** [1,†] **, Satoshi Iriyama** [2,†] **, Koki Jimbo** [2,†] **and Massimo Regoli** [3,*,†,‡]

[1]   Centro Vito Volterra, Via Columbia, 2, 00133 Roma, Italy; accardi@volterra.uniroma2.it
[2]   Department of Information Science, Tokyo University of Science, Yamazaki 2641, Japan;
     iriyama@is.noda.tus.ac.jp (S.I.); 7carmelooo@gmail.com (K.J.)
[3]   DICII, Engineering Faculty Via del Politecnico, Universitá di Roma Tor Vergata, 1, 00133 Roma, Italy
[*]   Correspondence: regoli@uniroma2.it
[†]   These authors contributed equally to this work.
[‡]   Current address: Viale del Politecnico, 2 00133 Roma, Italy.

**Abstract:**   A new class of public key agreement (PKA) algorithms called strongly-asymmetric algorithms (SAA) was introduced in a previous paper by some of the present authors. This class can be shown to include some of the best-known PKA algorithms, for example the Diffie–Hellman and several of its variants. In this paper, we construct a new version of the previous construction, called SAA-5, improving it in several points, as explained in the Introduction. In particular, the construction complexity is reduced, and at the same time, robustness is increased. Intuitively, the main difference between SAA-5 and the usual PKA consists of the fact that in the former class, *B* (Bob) has more than one public key and *A* (Alice) uses some of them to produce her public key and others to produce the secret shared key (SSK). This introduces an asymmetry between the sender of the message (*B*) and the receiver (*A*) and motivates the name for this class of algorithms. After describing the main steps of SAA-5, we discuss its breaking complexity assuming zero complexity of discrete logarithms and the computational complexity for both *A* and *B* to create SSK.

**Keywords:** public key exchange; cryptography; security

## 1. Introduction

PKA algorithms play an important role in the protection of privacy in IoT. However, the standard key length of usual PKA algorithms such as Diffie–Hellman or RSA [1,2], typically 512 bit, is not safe with respect to the eavesdropper's computational power [3]. On the other hand, increasing the key length also increases the computational complexity of the algorithm, thus decreasing its performance. After 40 more years since discovering the Diffie–Hellman, the study of modern PKA algorithms is widely spread in various mathematical fields. PKA based on using matrices have been considered in the literature, and they are based on the difficulty of solving a system of multivariate polynomial equations [4,5]. A generalized PKA class based on lattices was proposed in [6–9] and is a matrix-based cryptographic system, the attacks of which are reduced to the shortest vector problem. The application of PKA over rings introduces the new class of cryptographic systems, such as a fully-homomorphic encryption contributing to the development of a secure searchable encryption [10].

In the papers [11,12], a new scheme of public key agreement based on non-commutative algebra called a strongly-asymmetric public key agreement (SAPKA) was introduced. This scheme is very general, and in order to perform computational or estimate breaking complexity, concrete realizations are needed. Concrete realizations of the above-mentioned general scheme, called strongly-asymmetric algorithm 3 (SAA-3) and strongly-asymmetric algorithm 4 (SAA-4), were constructed in the [13].

The algorithms 3 (SAA-3) and 4 (SAA-4) are based on a public parameter $\alpha$, and in them, a receiver ($B$) is required to send a matrix basket to a sender ($A$) consisting of matrices commuting with one of his secret keys $x_B$. $A$ has to choose her secret key $x_A$ from this basket.

The present algorithm is an improved version of SAA-3 and SAA-4, called SAA-5, and the new points in it with respect to the previous ones are:

- The public parameter $\alpha$ is removed;
- All constraints on the secret keys of $B$ (see Conditions (1)–(4) in Section 1 of [13]) are reduced to the requirement that certain matrices should not be invertible;
- the attack developed in the remark after Equation (15) does not use commutativity assumptions;
- In [13], the secret key is a scalar. In SAA-5, this is replaced by a matrix, which makes exhaustive attacks impossible even in the case of low-dimensional matrices;
- all non-trivial constraints in the form of the secret key of $B$ are removed;
- In SAA-5, the way to construct and combine the public and secret keys of both $A$ and $B$ is different from [13];
- In SAA-5, $B$ does not need to send a matrix basket, thus decreasing the computational complexity;
- The important remark on the indeterminacy of the equations that condensate the attacker's information (see Theorem 2) is new.

Attacks are discussed in Section 4. The remark after Equation (15) explains the reason for the choice of the non-invertibility of some of the public keys of $B$.

Theorem 2 emphasizes another new feature of the present class of algorithms, namely that robustness against attacks is guaranteed not only by the difficulty of a problem, but also by its intrinsic indeterminacy: even if the attacker finds a solution, she still has to choose within a set of equivalent solutions obtained applying to it simple transformations. This set is so large so as to make exhaustive search impracticable.

An additional feature characterizing this class of algorithms is that the scheme on which they are based is not rigid, as in most PKA algorithms, but subject to an infinity of variations whose cryptographic merits are presently under investigation. An illustration of this statement is contained in the last section of the paper. In fact, for the simplicity of exposition, in all the previous sections, we have dealt with matrices with coefficients in a finite field (see the beginning of Section 2). However, looking at the proofs, it is clear that the whole construction works for matrices with entries in a ring $\mathcal{A}$ provided that there is the possibility of constructing invertible matrices with entries in $\mathcal{A}$. This possibility exists for a multiplicity of interesting rings, and in the numerical example discussed in Section 4.2, this situation is illustrated choosing $\mathcal{A} = \mathbb{Z}_{p-1}$ where $p$ is a prime number of order $2^{32}$. Even if the example is simple and low-dimensional ($d = 5$), exhaustive search is impracticable, and we have not been able to devise an alternative breaking strategy for it.

## 2. Steps of SAA-5

### 2.1. Public Parameters

The public parameters of the algorithm are:

- a natural integer $d \in \mathbb{N}$;
- a finite field $\mathbb{F}$ (typically $\mathbb{F} := \mathbb{Z}_p$, where $p$ is a large prime number);
- a finite set $I \subset \mathbb{N}$.

All scalar multiplications (in particular exponentiations) are meant in $\mathbb{F}$, and we use the convention:

$$0^x := 0 \quad ; \quad \forall x \in \mathbb{F}$$

The $d \times d$ matrices with entries in $\mathbb{F}$ are denoted $M(d; \mathbb{F})$, and the term *matrix* is used as a synonym of *element of* $M(d; \mathbb{F})$. Matrix multiplications are meant in the standard sense, while matrix

exponentiations are meant in the Schur sense, i.e., element-wise: if $c$ is either an element of $\mathbb{F}$ or a matrix $c = (c_{ij})$ and $M = (M_{a,b})$ is a matrix, the symbol $c^{\circ M}$ denoting the matrix:

$$\left(c^{\circ M}\right)_{a,b} := \begin{cases} c^{M_{a,b}} & \text{scalar case} \\ c_{a,b}^{M_{a,b}} & \text{matrix case} \end{cases} \quad ; \quad a,b \in \{1,\dots,d\}$$

is called *the Schur exponentiation of c by M*. Similarly, the Schur logarithm (in any basis) of a matrix $M$ is defined componentwise on the entries of $M$. Since in this paper, all logarithms considered are of the Schur-type, we simply write log to denote the Schur logarithm.

*2.2. Keys*

2.2.1. Secret Keys of $B$

They are matrices:

1. the main secret key of $B$:
$$x_B \in M(d;\mathbb{F})$$

2. additional secret keys of $B$:

$$\{A_j \in M(d;\mathbb{F}) \ : \ j \in I\} \quad ; \quad N_B \in M(d;\mathbb{F}) \quad ; \quad c \in M(d;\mathbb{F})$$

The only conditions to be satisfied by the secret keys of $B$ are:

- $N_B$ must be invertible;

- $c = c_0^{\circ c_1} =: c_0^{\circ \log c}$ with $\log c$ non Schur-invertible and:

$$c_{a,g} = c_{b,g} \quad ; \quad \forall a,b \tag{1}$$

- The $A_j$ ($j \in I$) are non-invertible (see the comments in Section 4).

2.2.2. Secret Key of $A$

$A$ chooses arbitrarily her secret key:

$$x_A \equiv (x_{A,j})_{j \in I} \quad ; \quad x_{A,j} \in M(d;\mathbb{F}) \, , \ \forall j \in I \tag{2}$$

2.2.3. SSK

The SSK is:
$$\kappa := c^{\circ(Q(x_A)x_B)}$$

where $Q \equiv (A_j)_{j \in I}$ is the linear map given by:

$$x \equiv (x_j)_{j \in I} \in M(d;\mathbb{F})^{|I|} \ \mapsto \ Q(x) := \sum_{j \in I} x_j A_j \in M(d;\mathbb{F})$$

where here and in the following, $|I|$ denotes the cardinality of the set $I$. Thus, the coefficients of $\kappa$ are:

$$\kappa_{a,g} := c^{[Q(x_A)x_B]_{a,g}} = \left(c^{\circ Q(x_A)x_B}\right)_{a,g} \quad ; \quad a,g \in \{1,\dots,d\}$$

### 2.2.4. Public Keys of *B*

The public keys of *B* are given by the finite set of matrices:

$$\{y_{B,2;j}\,,\, y_{B,3;j} \in M(d;\mathbb{F})\,:\, j \in I\}$$

constructed, using the secret keys of *B*, as follows.

For all $j \in I$ and $a,b \in \{1,\dots,d\}$:

$$y_{B,2;j;a,b} := c^{(A_j N_B)_{a,b}} = \left(c^{\circ A_j N_B}\right)_{a,b}$$

$$y_{B,3;j;a,b} := c^{(A_j x_B)_{a,b}} = \left(c^{\circ A_j x_B}\right)_{a,b}$$

### 2.2.5. Public Key of *A*

$$y_A := (y_{A;a,g}) \in M(d;\mathbb{F})\quad;\quad y_{A;a,g} = c^{[Q(x_A)N_B]_{a,g}} = \left(c^{\circ Q(x_A)N_B}\right)_{a,g}\,,\, a,g \in \{1,\dots,d\}$$

can be computed uniquely in terms of the public keys $(y_{B,2;j})$ of *B* and of the secret key of *A* as follows. For each $a,g \in \{1,\dots,d\}$, *A* computes:

$$y_{A;a,g} = \prod_{j \in I}\prod_{b \in \{1,\dots,d\}} (y_{B,2;j;b,g})^{(x_{A,j})_{a,b}} = \prod_{j \in I}\prod_{b \in \{1,\dots,d\}} \left(c^{(A_j N_B)_{b,g}}\right)^{(x_{A,j})_{a,b}}$$

$$= \prod_{j \in I}\prod_{b \in \{1,\dots,d\}} \left(c^{(x_{A,j})_{a,b}(A_j N_B)_{b,g}}\right) = c^{\sum_{j \in I}\sum_{b \in \{1,\dots,d\}}[x_{A,j}]_{a,b}(A_j N_B)_{b,g}} = c^{\sum_{j \in I}[x_{A,j}A_j N_B]_{a,g}}$$

$$= c^{[\sum_j x_{A,j}A_j N_B]_{a,g}} = c^{[Q(x_A)N_B]_{a,g}} = \left(c^{\circ Q(x_A)N_B}\right)_{a,g}$$

## 3. Protocol

*B* **computes the SSK** using the public key of *A* and his own secret keys.

**First step**: *B* uses his secret key $N_B$ to *clean the noise* calculating, for each $a,g \in \{1,\dots,d\}$:

$$\prod_{b \in \{1,\dots,d\}} (y_{A;a,b})^{(N_B^{-1})_{b,g}} = \prod_{b \in \{1,\dots,d\}} \left(c^{[Q(x_A)N_B]_{a,b}}\right)^{(N_B^{-1})_{b,g}}$$

$$= \prod_{b \in \{1,\dots,d\}} \left(c^{[Q(x_A)N_B]_{a,b}(N_B^{-1})_{b,g}}\right) = c^{\sum_b [Q(x_A)N_B]_{a,b}(N_B^{-1})_{b,g}}$$

$$= c^{([Q(x_A)N_B]N_B^{-1})_{a,g}} = c^{(Q(x_A))_{a,g}} = \left(c^{\circ Q(x_A)}\right)_{a,g} \tag{3}$$

**Second step**: Starting from (3), *B* inserts his main secret key calculating, for each $a,g \in \{1,\dots,d\}$:

$$\prod_{b \in \{1,\dots,d\}} \left(\left(c^{\circ Q(x_A)}\right)_{a,b}\right)^{(x_B)_{b,g}} = \prod_{b \in \{1,\dots,d\}} c^{Q(x_A)_{a,b}(x_B)_{b,g}} = c^{\sum_{b \in \{1,\dots,d\}} Q(x_A)_{a,b}(x_B)_{b,g}}$$

$$= c^{(Q(x_A)x_B)_{a,g}} = \left(c^{\circ Q(x_A)x_B}\right)_{a,g} = \kappa_{a,g}$$

Using the public keys $(y_{B,3;j})$ of $B$ and her own secret key, $A$ **computes the SSK** calculating, for each $a, g \in \{1, \ldots, d\}$:

$$\prod_{j \in I} \prod_{b \in \{1,\ldots,d\}} (y_{B,3;j;b,g})^{(x_{A,j})_{a,b}} = \prod_{j \in I} \prod_{b \in \{1,\ldots,d\}} (c^{(A_j x_B)_{b,g}})^{(x_{A,j})_{a,b}} = \prod_{j \in I} \prod_{b \in \{1,\ldots,d\}} (c^{(x_{A,j})_{a,b}(A_j x_B)_{b,g}})$$

$$= \prod_{j \in I} \prod_{b \in \{1,\ldots,d\}} c^{(x_{A,j})_{a,b}(A_j x_B)_{b,g}} = \prod_{j \in I} c^{\sum_b (x_{A,j})_{a,b}(A_j x_B)_{b,g}}$$

$$= \prod_{j \in I} c^{[x_{A,j} A_j x_B]_{a,g}} = c^{\sum_{j \in I} [x_{A,j} A_j x_B]_{a,g}} = \left( c^{\sum_{j \in I} x_{A,j} A_j x_B} \right)_{a,g} = \left( c^{Q(x_A) x_B} \right)_{a,g}$$

## 4. Attacks

In this section, we discuss the breaking complexity of the algorithm. We know that the eavesdropper ($E$) knows the public parameters, public keys, and the structure of public keys:

- $d, \mathbb{F}(p), I$
- $y_{B,2;j}, y_{B,3;j}$
- $y_A$

$E$ tries to recover the SSK:

$$\kappa_{a,g} = \left( c^{\circ Q(x_A) x_B} \right)_{a,g} \quad ; \quad a, g \in \{1, \ldots, d\}$$

In the following, all logarithms will be referred to a fixed, but arbitrary basis. Assuming zero cost logarithms, E computes for all $a, g \in \{1, \ldots, d\}$:

$$\log(y_A)_{a,g} = (Q(x_A) N_B)_{a,g} (\log c_{a,g})$$

$$\log(y_{B,2;j})_{a,g} = (A_j N_B)_{a,g} (\log c_{a,g})$$

$$\log(y_{B,3;j})_{a,g} = (A_j x_B)_{a,g} (\log c_{a,g})$$

Moreover, $E$ knows that:

$$\log(\kappa)_{a,g} = (Q(x_A) x_B)_{a,g} (\log c_{a,g})$$

In matrix notations and recalling that all logarithms are Schur logarithms, i.e., matrix logarithms are meant entry-wise:

$$\log y_A = (Q(x_A) N_B) \circ (\log c) \tag{4}$$

$$\log y_{B;2;j} = (A_j N_B) \circ (\log c) \quad ; \quad j \in I \tag{5}$$

$$\log y_{B;3;j} = (A_j x_B) \circ (\log c) \quad ; \quad j \in I \tag{6}$$

$$\log \kappa = (Q(x_A) x_B) \circ (\log c) \tag{7}$$

**Theorem 1.** *Suppose that:*
*(i)  for some $j \in I$, $A_j$ is invertible in the matrix sense,*
*(ii)  for the same $j$ as in (i), $\left( (\log c)^{\circ -1} \circ \log y_{B;2;j} \right)$ is invertible in the matrix sense,*
*(iii)  $\log c$ is Schur-invertible.*
*Then, the SSK satisfies the equation:*

$$\log \kappa = \left( \left( (\log c)^{\circ -1} \circ \log y_A \right) \left( (\log c)^{\circ -1} \circ \log y_{B;2;j} \right)^{-1} \left( \log y_{B;3;j} \circ (\log c)^{\circ -1} \right) \right) \circ (\log c) \tag{8}$$

*where $(\log c)^{\circ -1}$ denotes the Schur inverse of $\log c$.*

**Remark 1.** *Since $N_B$ is matrix-invertible by assumption, Condition (i) implies that the product $A_j N_B$ is matrix-invertible. However, the product of a matrix-invertible and a Schur-invertible matrix need not be matrix-invertible. Therefore Assumption (ii) is necessary for the proof of (8).*

**Proof.** Since by Assumption (iii) $\log c$ is Schur-invertible, (4) is equivalent to:

$$\log y_A \circ (\log c)^{\circ -1} = Q(x_A) N_B \iff \left( (\log c)^{\circ -1} \circ \log y_A \right) N_B^{-1} = Q(x_A)$$

Under Assumptions (i) and (ii), (5) is equivalent to:

$$\log y_{B;2;j} \circ (\log c)^{\circ -1} = A_j N_B \iff A_j^{-1} \left( (\log c)^{\circ -1} \circ \log y_{B;2;j} \right) = N_B$$

$$\iff \left( (\log c)^{\circ -1} \circ \log y_{B;2;j} \right)^{-1} A_j = N_B^{-1}$$

and combining the two results:

$$Q(x_A) = \left( (\log c)^{\circ -1} \circ \log y_A \right) \left( \left( (\log c)^{\circ -1} \circ \log y_{B;2;j} \right)^{-1} A_j \right)$$

$$= \left( (\log c)^{\circ -1} \circ \log y_A \right) \left( (\log c)^{\circ -1} \circ \log y_{B;2;j} \right)^{-1} A_j$$

Finally, from (6) and Assumption (i), we get:

$$\log y_{B;3;j} \circ (\log c)^{\circ -1} = A_j x_B \iff A_j^{-1} \left( \log y_{B;3;j} \circ (\log c)^{\circ -1} \right) = x_B$$

Inserting in (7) these two results, one gets

$$\log \kappa = \left( \left( \left( (\log c)^{\circ -1} \circ \log y_A \right) \left( (\log c)^{\circ -1} \circ \log y_{B;2;j} \right)^{-1} A_j \right) \left( A_j^{-1} \left( \log y_{B;3;j} \circ (\log c)^{\circ -1} \right) \right) \right) \circ (\log c)$$
$$\left( \left( (\log c)^{\circ -1} \circ \log y_A \right) \left( (\log c)^{\circ -1} \circ \log y_{B;2;j} \right)^{-1} A_j A_j^{-1} \left( \log y_{B;3;j} \circ (\log c)^{\circ -1} \right) \right) \circ (\log c)$$
$$\left( \left( (\log c)^{\circ -1} \circ \log y_A \right) \left( (\log c)^{\circ -1} \circ \log y_{B;2;j} \right)^{-1} \left( \log y_{B;3;j} \circ (\log c)^{\circ -1} \right) \right) \circ (\log c)$$

which is (8). □

**Corollary 1.** *In the assumptions of Theorem 1, suppose that the Schur products in (8) coincide with the matrix products. Then, the SSK satisfies the equation:*

$$\log \kappa = (\log c)^{-1} \log y_A \log y_{B;2;j}^{-1} (\log c) \log y_{B;3;j} \tag{9}$$

**Proof.** Under the assumptions of the corollary, (8) becomes:

$$\log \kappa = \left( \left( (\log c)^{\circ -1} \circ \log y_A \right) \left( (\log c)^{\circ -1} \circ \log y_{B;2;j} \right)^{-1} \left( \log y_{B;3;j} \circ (\log c)^{\circ -1} \right) \right) \circ (\log c)$$

$$= (\log c)^{-1} \log y_A \left( (\log c)^{-1} \log y_{B;2;j} \right)^{-1} \log y_{B;3;j} (\log c)^{-1} \log c$$

$$= (\log c)^{-1} \log y_A \log y_{B;2;j}^{-1} (\log c) \log y_{B;3;j} (\log c)^{-1} \log c$$

$$= (\log c)^{-1} \log y_A \log y_{B;2;j}^{-1} (\log c) \log y_{B;3;j}$$

□

**Corollary 2.** *If the conditions of both Theorem 1 and Corollary 1 are satisfied and, in addition, $(\log c)$ commutes with $\log y_A \log y_{B;2;j}^{-1}$, then:*

$$\log \kappa = \log y_A \log y_{B;2;j}^{-1} \log y_{B;3;j} \tag{10}$$

**Proof.** Under the given assumptions, Equation (9) becomes:

$$\log \kappa = (\log c)^{-1} \log y_A \log y_{B;2;j}^{-1} (\log c) \log y_{B;3;j} = \log y_A \log y_{B;2;j}^{-1} \log y_{B;3;j}$$

which is (10)　□

**Remark 2.** *If $\log c$ is a scalar $\neq 0$, Condition (iii) of Theorem 1 and the conditions of Corollary 1 and Corollary 2 are automatically satisfied. Therefore, in this case, under Conditions (i) and (ii) of Theorem 1, Equation (10) says that the SSK is a function of the public parameters, i.e., the algorithm is breakable. However, it is easy for B to construct his secret keys so that either Condition (i) or (ii) of Theorem 1 is violated. For example, B can choose all the $A_j$ ($j \in I$) so that they are not matrix-invertible, thus violating Condition (ii).*
　*If $\log c$ is a matrix, it is sufficient that it has a single zero entry to violate Condition (iii) of Theorem 1.*

Equations (4)–(7) are $2 + 2|I|$ cubic matrix equations depending on the $5 + |I|$ matrix unknowns:

$$Q(x_A) =: x_1 \; ; \; N_B =: x_2 \; ; \; A_j =: x_{3;j} \; ; \; x_B =: x_4 \; ; \; \log \kappa =: x_5 \, , \, \log c$$

where the left-hand sides of Equations (4)–(6) are known to $E$:

$$\alpha_1 := \log y_A \quad ; \quad \alpha_{2;j} := \log y_{B,2;j} \quad ; \quad \alpha_{3;j} := \log y_{B,3;j} \tag{11}$$

With these notations, $E$ finds the system of cubic equations:

$$\alpha_1 = (x_1 x_2) \circ \log c \tag{12}$$

$$\alpha_{2;j} = (x_{3;j} x_2) \circ \log c \; ; \quad j \in I \tag{13}$$

$$\alpha_{3;j} = (x_{3;j} x_4) \circ \log c \; ; \quad j \in I \tag{14}$$

$$x_5 = (x_1 x_4) \circ \log c \tag{15}$$

from which she wants to derive $x_5$.

**Remark 3.** *Theorem 1 explains why it is convenient to choose the $x_{3;j} = A_j$ not invertible for all $j \in I$ and why it is convenient to choose $\log c$ to be a non-Schur-invertible matrix.*

In fact, in this case, the direct attack to the SSK of Theorem 1 is not applicable, and $E$ faces the problem of solving the cubic system given by Equations (12)–(15). Since the cubic non-linearity is given by matrix multiplication, the scalar unknowns are strongly entangled, and it is known that this brings the complexity of the application of Groebner-type algorithms near the upper bound, which is super-exponential.

In addition to this, there is another more substantial difficulty for $E$ given by the fact that, as shown by the following theorem, the above-mentioned system is **intrinsically indeterminate**.

**Theorem 2.** *Suppose that $(x_1, x_2, (x_{3,j})_{j \in I}, x_4, \log c, x_5)$ is a solution of the system (12)–(15). Then, for any pair $(u, v)$ of invertible $d \times d$ matrices, $(x_1 u^{-1}, u x_2 v^{-1}, (x_{3,j} u^{-1})_{j \in I}, u x_4 v^{-1}, v \log c, x_5)$ is a solution of the same system.*

**Proof.** It is sufficient to prove that the change of variables:

$$\begin{cases} x_1 \to x_1 u^{-1} \\ x_2 \to u x_2 \\ x_{3,j} \to x_{3,j} u^{-1} \\ x_4 \to u x_4 \end{cases}$$

leaves the right-hand sides of Equations (12)–(15) unaltered. In fact:

$$(x_1 x_2) \circ \log c \to (x_1 u^{-1} u x_2) \circ \log c = (x_1 x_2) \circ \log c$$

$$(x_{3;j} x_2) \circ \log c \to (x_{3,j} u^{-1} u x_2) \circ \log c = (x_{3;j} x_2) \circ \log c$$

$$(x_{3;j} x_4) \circ \log c \to (x_{3,j} u^{-1} u x_4) \circ \log c = (x_{3;j} x_4) \circ \log c$$

$$(x_1 x_4) \circ \log c \to (x_1 u^{-1} u x_4) \circ \log c = (x_1 x_4) \circ \log c$$

□

An additional indeterminacy, with respect to the one described by Theorem 1, is the one arising in Equations (13) and (14) from the non-invertibility of $A_j = x_{3;j}$. However, even neglecting this one, Theorem 2 means that, even if $E$ finds a solution of the system (12)–(15), she has to choose among all the solutions obtained from it applying the transformations described in Theorem 2. Exhaustive search among these solutions, which are equi-probable for $E$ given her level of information, are impracticable even for $\mathbb{F} = \mathbb{Z}_p$ with $p$ a relatively small prime (say of the order $2^{32}$) because their cardinality is of the same order as the cardinality of $M(d; \mathbb{F})$.

*4.1. Computational Complexity*

The computational complexity for A is given by:

- computation of $y_A$
- computation of the SSK

In the computation of $y_A$, A computes for each element $a, g \in \{1, \dots, d\}$:

$$y_{A;a,g} = \prod_{j \in I} \prod_{b \in \{1,\dots,d\}} (y_{B,2;j;b,g})^{(x_{A,j})_{a,b}}$$

The number of total scalar exponentiations is:

$$\text{exponentiations: } d^3 |I|$$

and the number of total scalar multiplications is:

$$\text{scalar multiplication: } d^2 (d-1)(|I|-1)$$

A computes the SSK as:

$$(\kappa)_{a,g} = \prod_{j \in I} \prod_{b \in \{1,\dots,d\}} (y_{B,3;j;b,g})^{(x_{A,j})_{a,b}}$$

The number of total scalar exponentiations is:

$$\text{exponentiations: } d^3 |I|$$

the number of total scalar multiplications is:

$$\text{scalar multiplication: } d^2(d-1)(|I|-1)$$

Therefore, the total number of exponentiations is:

$$2d^3|I| \sim d^3|I|$$

The total number of scalar multiplications is:

$$2d^2(d-1)(|I|-1) \sim d^3|I|$$

The computational complexity for B is given by:

- computation of $y_{B,2;j}$
- computation of $y_{B,3;j}$
- computation of the SSK

The calculation of each $(y_{B,2;j})_{a,b} = c^{(A_j N_B)_{a,b}}$ or $(y_{B,3;j})_{a,b} = c^{(A_j x_B)_{a,b}}$ requires $d^2|I|$ scalar exponentiations and $|I|$ matrix products.

The calculation for SSK has two parts. The first part is the calculation of:

$$(\kappa')_{a,g} = \prod_{b \in \{1,\dots,d\}} (y_{A;a,b})^{(N_B^{-1})_{b,g}}$$

The second part is given by:

$$(\kappa)_{a,g} = \prod_{b \in \{1,\dots,d\}} \left((\kappa')_{a,b}\right)^{(x_B)_{b,g}}$$

Each part contains $d^3$ scalar exponentiations and $d^2(d-1)$ scalar multiplications. Therefore, the total number of scalar exponentiations is:

$$2d^2|I| + 2d^3 \sim d^2(d+|I|)$$

The total number of scalar multiplications is:

$$2d^2(d-1) \sim d^3$$

The total number of matrix multiplications is:

$$2|I| \sim |I|$$

*4.2. A Numerical Example*

Here, we construct a numerical example of SAA5. The setting is the following:

- $d = 5$: (dimension)
- $\mathbb{F} = \mathbb{Z}_p$
- $p = 4294967291$: a prime number, $2^{31} < p < 2^{32}$
- $c = 1234567891$: a prime number such that $g.c.d(c,p) = 1$
- $I = \{1,2,3\}$: a set, $|I| = 3$

Since $g.c.d(c,p) = 1$, the parameter $c$ has period $p-1$. Therefore, the function $f_{c,p}(x) = c^x$ is periodic with period $p-1$. Therefore, to avoid large numbers and keep the computations within the 32-bit domain, it is convenient to perform all operations that involve exponents of $c$ modulo $p-1$, while the multiplication of exponentials should be performed modulo $p$.

To avoid the use of a double module, we choose the coefficients of all secret keys in $\mathbb{Z}_{p-1}$, and all operations are made in this ring. This has also the advantage that, since $\mathbb{Z}_{p-1}$ is a ring and not a field, all invertibility issues become more difficult in this framework. This fact will be exploited in greater generality in a future publication.

Bob chooses the secret key $x_B \in M(d, \mathbb{Z}_{p-1})$ as:

$$
x_{b,3} = \begin{pmatrix}
1302223311 & 3036102706 & 1950911555 & 1588574439 & 4205392019 \\
475199933 & 1588871204 & 3380984642 & 2028686256 & 1410372785 \\
84237198 & 331418214 & 377622969 & 94920131 & 1897882575 \\
1264609458 & 2047942517 & 2633489909 & 2475676273 & 3402425250 \\
2609452388 & 906983806 & 438577591 & 1027462714 & 922571658
\end{pmatrix}
$$

and additional secret keys:

$$
A_1 = \begin{pmatrix}
0 & 3788905542 & 2222785513 & 2073815497 & 3603745241 \\
3105694558 & 0 & 4265220769 & 1017644232 & 3445055000 \\
2572706715 & 3038815660 & 0 & 3146135139 & 4131986064 \\
1041303085 & 1727576570 & 148785315 & 0 & 3474561565 \\
338748084 & 3560001775 & 248998463 & 2247690485 & 0
\end{pmatrix}
$$

$$
A_2 = \begin{pmatrix}
0 & 4188102984 & 4261217150 & 3173675344 & 3453829986 \\
2184532822 & 0 & 3044136815 & 1979293276 & 2532458993 \\
157875694 & 1185021703 & 0 & 2332943479 & 629241533 \\
3487714523 & 4162196921 & 3314231639 & 0 & 1094496175 \\
202417627 & 2856466936 & 1387021057 & 389944647 & 0
\end{pmatrix}
$$

$$
A_3 = \begin{pmatrix}
0 & 3301308627 & 3171841047 & 1901994323 & 3927695075 \\
893545770 & 0 & 1019361737 & 3252355315 & 1995919988 \\
79669926 & 3071982 & 0 & 4219791915 & 1400948265 \\
4212083462 & 3039196150 & 3154904801 & 0 & 1562880115 \\
2140484588 & 4129219179 & 1060549870 & 4167897854 & 0
\end{pmatrix}
$$

$$
N_B = \begin{pmatrix}
1914362363 & 1917061405 & 1176808279 & 648676785 & 1764822838 \\
4212500437 & 493998612 & 1018470348 & 659200518 & 4041739862 \\
2425394518 & 1010843968 & 1551629103 & 3916843822 & 4115443575 \\
480122131 & 3641672291 & 2759527160 & 1523231740 & 3385651728 \\
3402493211 & 3829823144 & 1677835446 & 3072388584 & 1656686133
\end{pmatrix}
$$

$N_B$ is invertible in $\mathbb{Z}_{p-1}$, and $N_B^{-1}$ is calculated as:

$$
N_B^{-1} = \begin{pmatrix}
1056670352 & 1752780917 & 1091186792 & 249921880 & 484074830 \\
2519741046 & 3121847791 & 2664334830 & 155918887 & 2564514478 \\
2875916262 & 676103283 & 1301763609 & 2178347744 & 1367832327 \\
4158981885 & 2063040371 & 953376899 & 2272312361 & 1874512565 \\
2507129114 & 981826659 & 1817535756 & 699581298 & 2543768509
\end{pmatrix}
$$

In fact,

$$
N_B N_B^{-1} = I \bmod p - 1
$$

Bob calculates the public keys $y_{B,2,j}$ and $y_{B,3,j}$ for $j \in I$:

$$
y_{B,2;1} = \begin{pmatrix}
520755896 & 2629144795 & 598609158 & 3930924878 & 565565295 \\
1005048396 & 136057902 & 803662542 & 3450162971 & 1782017006 \\
313572865 & 2862336142 & 532367644 & 2658869746 & 1063794269 \\
1133278001 & 514281167 & 3782874102 & 1501107275 & 2291906133 \\
3747999327 & 958748864 & 4039733998 & 3623773602 & 221441433
\end{pmatrix}
$$

$$
y_{B,2;2} = \begin{pmatrix}
1100675738 & 2377306807 & 320310045 & 4183872877 & 3094588329 \\
1858941337 & 2590258583 & 3019711351 & 504050841 & 3545977310 \\
871154770 & 3112430422 & 2991114005 & 2307732604 & 1533414277 \\
2575749561 & 2383342108 & 3906470091 & 1327027748 & 1942980649 \\
4177738145 & 351928788 & 3414324786 & 2101677812 & 3606279630
\end{pmatrix}
$$

$$
y_{B,2;3} = \begin{pmatrix}
1211352515 & 2267569344 & 1018962528 & 3323274536 & 1724991904 \\
1276774275 & 972135244 & 799382061 & 2659295349 & 2254929026 \\
595742140 & 3824016846 & 3562771239 & 2559354870 & 557977523 \\
1828792733 & 3028675089 & 1029846831 & 445923660 & 1942381250 \\
776947273 & 3890081018 & 1690304752 & 3535197303 & 3228614369
\end{pmatrix}
$$

$$
y_{B,3;1} = \begin{pmatrix}
3192346363 & 3313915884 & 148849609 & 2067376731 & 1040074624 \\
3767002677 & 4002437607 & 1213185627 & 2779519475 & 2749545589 \\
1981484667 & 2883640587 & 1349676569 & 3826297559 & 1250751365 \\
3988510025 & 924511781 & 3273249165 & 3498236201 & 4112494691 \\
2523966937 & 239944636 & 1780986054 & 2452246615 & 139198803
\end{pmatrix}
$$

$$
y_{B,3;2} = \begin{pmatrix}
3487972825 & 2059946345 & 4149166921 & 1468549539 & 547514569 \\
2888355633 & 501003547 & 985587578 & 3425801686 & 2614575952 \\
1865561822 & 2914838138 & 3935514555 & 1461994052 & 2528354877 \\
2466289308 & 2410324226 & 622706326 & 1831851213 & 502543542 \\
223847409 & 4254497091 & 2741822882 & 1862234570 & 4293984694
\end{pmatrix}
$$

$$
y_{B,3;3} = \begin{pmatrix}
1887865820 & 105467147 & 3788887540 & 769774167 & 1287602684 \\
571995438 & 620356231 & 2831953178 & 3695422732 & 1184503561 \\
834453900 & 1318534845 & 830504770 & 3499684766 & 2702832283 \\
3297267532 & 3289783332 & 2224144356 & 978328852 & 3590377270 \\
1613472801 & 3559725977 & 1198572164 & 728404861 & 2448481277
\end{pmatrix}
$$

Alice calculates the public key $y_A$ after receiving Bob's public keys $y_{B,2,j}$:

$$
y_A = \begin{pmatrix}
2654282219 & 3218104680 & 1840335336 & 281612527 & 871286734 \\
3758875123 & 3123626985 & 1756470990 & 3091679784 & 3513893738 \\
1605723039 & 2231283615 & 3496004106 & 51747848 & 2854303327 \\
4057281561 & 3744174842 & 2830691803 & 2886194642 & 1545723227 \\
3119611268 & 507195559 & 901862328 & 81086314 & 2422636784
\end{pmatrix}
$$

In order to obtain the SSK, Bob first calculates $x'_B$:

$$
x'_B = \begin{pmatrix}
1568885684 & 464596335 & 2461272026 & 449127471 & 1783528355 \\
205414769 & 1535631617 & 1239746722 & 3825791910 & 317352834 \\
253683714 & 144235286 & 3154427854 & 2969897183 & 3320441305 \\
559112875 & 2542680230 & 3384129982 & 3734459150 & 3827314646 \\
1283770816 & 2487602481 & 2851739705 & 3932446652 & 2790935753
\end{pmatrix}
$$

Finally, Bob calculates his SSK $k_B^{(SSK)}$ using Alice's public key $y_A$:

$$k_B^{(SSK)} = \begin{pmatrix} 4118803775 & 3024367129 & 2201420160 & 2335335312 & 46065376 \\ 1384844995 & 607556554 & 2645672430 & 4136350896 & 3596845616 \\ 4209215563 & 1529533803 & 1525531379 & 781854571 & 2723231816 \\ 1625920071 & 3671248796 & 1470525740 & 3884958370 & 1972389092 \\ 2062666758 & 774480666 & 1689604710 & 2098990694 & 1929943712 \end{pmatrix}$$

Alice calculates the SSK $k_A^{(SSK)}$ using public key $y_{B,3,j}$ as:

$$k_A^{(SSK)} = \begin{pmatrix} 4118803775 & 3024367129 & 2201420160 & 2335335312 & 46065376 \\ 1384844995 & 607556554 & 2645672430 & 4136350896 & 3596845616 \\ 4209215563 & 1529533803 & 1525531379 & 781854571 & 2723231816 \\ 1625920071 & 3671248796 & 1470525740 & 3884958370 & 1972389092 \\ 2062666758 & 774480666 & 1689604710 & 2098990694 & 1929943712 \end{pmatrix}$$

One can check that both secret keys $k_B^{(SSK)}$ and $k_A^{(SSK)}$ are same.

**Author Contributions:** Conceptualization, L.A., S.I., M.R., K.J.; software, M.R.; validation, K.J.; formal analysis, S.I., M.R. and K.J.; data curation, S.I.; writing—original draft preparation, S.I., K.J.; writing—review and editing, L.A.; supervision, L.A.

**Funding:** This research received no external funding.

**Acknowledgments:** A special thanks goes to the memory of Masanori Ohya, friend, "Maestro" and colleague.

**Conflicts of Interest:** The authors declare no conflict of interest.

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
