# Peer review of "A New Class of Strongly Asymmetric PKA Algorithms: SAA-5"

_cryptography, doi:10.3390/cryptography3010009_

Round 1
Reviewer 1 Report
This paper is not suitable for publication in its current form. Its looks like a patent disclosure quickly turned into a paper.
1- The abstract said: "the present paper is a continuation of the program initiated in [4] .....": it is not acceptable to expect the readers to read different documents to be able to understand this paper. The authors should summarize prior work as part of the introduction, and background information. In general authors should not include references in the abstract.
2- The introduction is not a real introduction, is is a list of what the paper cover. The authors should provide a step by step description of the area of interest, including references. This paper has only 4 references, mainly originating from the authors. The authors need to convey their understanding of what other did in the field.
3- The concept of a matrix of private/public key pairs is extremely valuable. The authors are doing a decent job to describe the scheme, and mathematics. This is innovative, and interesting. Some of the terms need additional information. For example define NB,3 line 56.
4- The use of this technology for public key infrastructure is not clear. The authors need to explain how a certificate authority can track the key distribution, and authenticate the users. Can a man-in-the-middle attack be effective? What prevents third party to generate a public-private key pair, and pretends to be Bob?
In brief, this is an interesting work, but the authors need to turning it into a quality paper. As is this should not be published.
Author Response
Dear Editors
We express appreciation to the referees giving us the fruitful comments. According to them, we revised the article carefully.
The points that we modified are the following:
- In the abstract references have been suppressed
- The proof of the last statement is not included in the paper because it doesn't add much to the original contents of the paper and any expert reader can easily verify this statement.
- All symbols $x_{B,3}$ and $N_{B,3}$ have been replaced respectively by $x_{B}$ and $N_{B}$. The $3$ x index referred to a previous version of the algorithm.
- The introduction is updated. We mentioned the other works related to our study, and some applications using matric based PKA, and explained the difference between the previous versions and present one.
- That we added condition (1)
- That in section 4 we introduced a new notation to distinguish between Schur multiplication and matrix multiplication.
- That we have formulated the various steps of the attacks as formal theorems, Lemmas, Corollaries rather than as Remarks as before, in order to make more clear the assumptions needed for each single attack.
- We added 6 more references by other groups related to our article.
best regards,
Luigi Accardi
Satoshi Iriyama
Koki Jimbo
Massimo Regoli
2019, 1st of March
Reviewer 2 Report
I would like to see additional relevant references, if possible.

Author Response

(The authors gave the same response as above.)

Reviewer 3 Report
The style of this paper is one definitely specialized for Cryptography and its unique brand of reader. The content is very relevant the goals of the journal. The paper offers a strong extension to prior work by the authors. The soundness of the proposed encryption/decryption method is supported by both general mathematical argument and specific examples. I support the publication of this paper.